# Typomorphic Characteristics of Gold-Bearing Pyrite and Its Genetic Implications for the Fang'an Gold Deposit, the Bengbu Uplift, Eastern China

Ying Wang [1], Li Xiong [2], Ze Zhong [1], Shenglian Ren [1,*], Gang Zhang [1], Juan Wang [1], Yan Zhang [1] and Chuanzhong Song [1]

[1] School of Resources and Environmental Engineering, Hefei University of Technology, Hefei 230009, China
[2] Jiangxi Disaster Reduction and Preparedness Center, Nanchang 330000, China
* Correspondence: 1999800079@hfut.edu.cn; Tel.: +86-13605512166

**Abstract:** The Fang'an quartz-vein gold deposit is located in the eastern part of the Bengbu uplift. The eastern part of the Bengbu uplift is considered to be the western extension of the Zhaoyuan gold mineralization zone in the Jiaodong area of Shandong Province and has huge mineralization potential. The Fang'an deposit was a newly discovered, small-sized gold deposit, and the research in the area is relatively weak. In this study, samples of quartz-vein type ore were collected from the ZK141, ZK1549, and ZK1665 drill holes of the Fang'an gold deposit. Based on the study of the geological characteristics, the major and trace elements of pyrite in different stages were analyzed by electron probe microanalyzer (EPMA), to explore the compositional characteristics of pyrite, the occurrence of gold, and the source of ore-forming fluid. The studies indicate that the deposit experienced four ore-forming stages: the quartz stage, the quartz-pyrite stage, the polymetallic sulfide stage, and the carbonate stage. The pyrites are grouped into three stages, corresponding to the first three ore-forming stages. The EPMA analyses showed that the major elements of pyrite were high Fe and low S, indicating that the formation was hydrothermal. The high content of Ni indicated that the metallogenic materials were derived from between the mantle and the lower crust. The general Co/Ni ratio of >1, with an average of 5.34, indicated that the ore-forming fluid was derived from the magmatic–hydrothermal fluid and wall rock. The Fang'an gold deposit mainly contains nano-gold.

**Keywords:** Fang'an gold deposit; pyrite; EPMA; compositional characteristics; nano-gold




## 1. Introduction

The Bengbu uplift is located in the southeastern margin of the North China plate (NCP) and is bound by the Tan-Lu fault to the east and by the Hefei basin and the Dabie orogenic belt to the south, respectively (Figure 1a) [1–3]. It is distributed as a nearly EW-trending belt. The Bengbu uplift is marked by ductile and brittle–ductile tectonics and magmatic activity, which is favorable for gold mineralization. Thus far, many gold deposits have been found [4–9], such as Dagongshan, Mao shan, Rongdu, Zhu Ding, etc. In recent years, some researchers have conducted detailed studies on quartz-vein type gold deposits (the Maoshan and Dagongshan and Rongdu deposits) in the Bengbu uplift, including their ore-forming age, ore-forming fluid, source material, and metallogenic mechanisms [4–14]. The ore-forming age was mainly at 110~160 Ma [8–10,12]. The ore-forming fluid is mainly metamorphic fluid or magmatic fluid [4,5,7–9]. The metallogenic materials are divided from the metamorphic basement, the mantle source material, or the magmatic rocks [10–14]. Based on several features of the gold deposits in the Bengbu uplift, including their occurrence within the Precambrian metamorphic terrane [11,12], their Mesozoic formation age, the predominance of low salinity $H_2O$-$CO_2$-NaCl fluids [10], and the gold-related magmatism (115–130 Ma), which is coeval with the gold mineralization (113–134 Ma) [5], geologists classified these deposits as magmatic–hydrothermal or orogenic

gold deposits. These deposits have a similar geological background, but a different genesis. Therefore, the key problem of the genesis of the deposit has not been solved.

**Figure 1.** (**a**) Location of Figure 1b in China; (**b**) tectonic location of Bengbu uplift in eastern China; (**c**) geological map of Fang'an gold deposit, modified after [15].

The Fang'an deposit is a small-sized, quartz-vein-type gold deposit, which is located in the east of the Bengbu uplift. It is a new breakthrough of the prospecting work of the No.312 Geological Team, Anhui Bureau of Geology and Mineral Resources. However, research on the Fang'an deposit is still limited, and the material source and genesis of the gold deposit need further study. According to the content and distribution characteristics of the major and trace elements of pyrite, this paper selected pyrite in the Fang'an gold deposit for mineralogy and EPMA analysis. Through the study of the typomorphic characteristics of pyrite, the genesis of the deposit and the change of the mineralization environment were revealed, providing evidence for the genesis of the deposit and the source of the gold material.

## 2. Geology Characteristics

### 2.1. Regional Geology

The Bengbu uplift is located in the Bengbu-Fengyang-Wuhe area in northern Anhui Province and is tectonically located in the southeastern North China plate, at the intersection of the North China plate and the Yangtze block and to the west of the Tan-Lu fault. It generally has an east–west distribution (Figure 1b). The majority of the Bengbu uplift is covered by the Quaternary, and the exposed metamorphic basement is mainly the Archaean Wuhe group (Ar3wh). The Wuhe group is composed of five subgroup formations: Xigudui, Zhuangzili, Fengshanli, Xiaozhangzhuang, and Yinjiajian [16]. The protolith of the Wuhe group is a series of basic-intermediate-felsic volcanic rocks, containing graywacke and iron siliciclastic carbonate rock, which are in contact with the near subhorizontal ductile shear zone. Granitoids are widely exposed in the Bengbu uplift, mainly granite, monzogranite,

granodiorite, and moyite [17], which are in intrusive contact with the wall rocks (Wuhe group). Yang et al. (2005, 2006) reported that the age of the magmatic intrusion is Paleoproterozoic (2.1 Ga) by zircon U-Pb of the moyite from the Bengbu uplift [18,19]. Wang (2012) suggested that the moyite is part of the metamorphic basement of the Wuhe group [20]. Moreover, late granitic intrusions, including granite, monzogranite, and granodiorite, were emplaced into the metamorphic basement during the Mesozoic (160~110 Ma) [18,21–23]. The structures mainly comprise E- and NE- to NNE-trending faults in the Bengbu uplift, which constitute the tectonic pattern of the region. Furthermore, the NE- to NNE-trending structures are thought to be subsidiary to the regional Tan-Lu fault.

The Fang'an gold deposit is located in the east of the Bengbu uplift and is an important junction of the EW-trending Bengbu uplift and NNE-trending Tan-Lu fault (Figure 1c) [24]. The Fang'an deposit is a quartz-vein-type gold deposit controlled by the Xigudui formation and the NNE-trending fault structure.

*2.2. Deposit Geology*

The Fang'an gold deposit is mainly covered by the Quaternary, with a depth of about 60.33~90.70 m. The proven reserves of gold are 479 kg, with an average grade of 2.63 g/t. According to the drilling date, the metamorphic basement in the area is mainly composed of the Xigudui formation of the Wuhe group ($Ar_3$-$Pt_1w$). The Xigudui formation is mainly amphibolite, amphibole plagioclase gneiss, and biotite plagiogneiss. It has undergone amphibolite facies metamorphism.

The Fang'an gold deposit is located at the junction of the Wuhe-Hongxinpu fault zone and the Zhuding–Shimenshan fault zone (Figure 1c) [24]. Both are secondary fractures of the Tan-Lu fault. The Fang'an gold deposit consists of two metallogenic belts, I and II. The ore-bearing structures are ductile–brittle shear belts; the strike is nearly NS-trending and the dip nearly SE-trending (Figure 2a).

The magmatic rocks are mainly granitic porphyry, diorite porphyry, diorite, and lamprophyre in the ore field. The Nvshan intrusion is located 6 km south of the ore field, and the formation age of the megaporphyritic biotite granite is 130.1 ± 3.2 Ma. This indicates that the megaporphyritic biotite granite was formed in the early Cretaceous magmatic event [18].

The lenticular orebodies are controlled by the brittle–ductile fracture zone. They strike NNE-trending and dip to SEE at 30–60°, and the ductile fracture zone is almost parallel to the Tan-Lu fault. The orebodies show pinch-out or reappearance within the ore-bearing fractures. Gold-bearing quartz veins and the gold ore body are primarily hosted in sericitic quartzite and amphibolite.

The ore minerals mainly occur in the polymetallic sulfide quartz vein and are mainly pyrites, with a small amount of chalcopyrite, sphalerite, and galena (Figures 3 and 4). The gangue minerals are quartz, barite, and dolomite. The alteration types of the Fang'an gold deposit are pyritization (Figure 3a,b), silicification (Figure 3c), sericitization (Figures 3d and 4a), and carbonation (Figure 3d), showing the medium-low temperature hydrothermal alteration combination. The early stage minerals are often crosscut and superimposed by late-stage minerals, resulting in mineral assemblages at different stages.

Based on the observation of the drill cores and combined with microscopic analysis, the mineral assemblage, structures, and crosscutting relationships between the ore veins were studied. The mineralization process of the Fang'an gold deposit can be divided into four stages, including the quartz stage (I), where the mineral assemblage is quartz, sericite (Figure 4a) and a small amount of coarse-grained structured pyrite (Figure 4b), mainly in the form of milky white quartz veins penetrating the wall rocks. In the quartz-pyrite stage (II), the mineral assemblage is quartz, sericite, and pyrite, mainly in the form of a vein structure through the wall rock. Pyrite is a crushing phenomenon (Figure 4c). The Py2 occurs by brecciation, the subsequent replacement by chalcopyrite, and rimming by Py3 (Figure 4d). In the polymetallic sulfide stage (III), the mineral assemblage is quartz, pyrite, galena, and sphalerite, associated with a small amount of chalcopyrite. The ore texture is

mainly granular and metasomatic. Galena, sphalerite, and chalcopyrite occur in this stage as anhedral phases infilling porous assemblages or as anhedral grains occluding the space between the earlier, variable CL-zoned prismatic quartz. The main ore structures include disseminated and vein (Figure 4d,e). In the carbonate stage (IV), the mineral assemblage is calcite, quartz, and sericite and shows veinlet forms cutting the first three stages (Figure 4f). The pyrite is divided into three stages, showing the Py1–Py3 symbol, corresponding to the first three ore-forming stages. The mineral paragenetic assemblage is shown in Figure 5. Due to pyrite widely occurring in the ore-forming stage, research on its composition is beneficial for revealing the environmental changes of the ore-forming process.

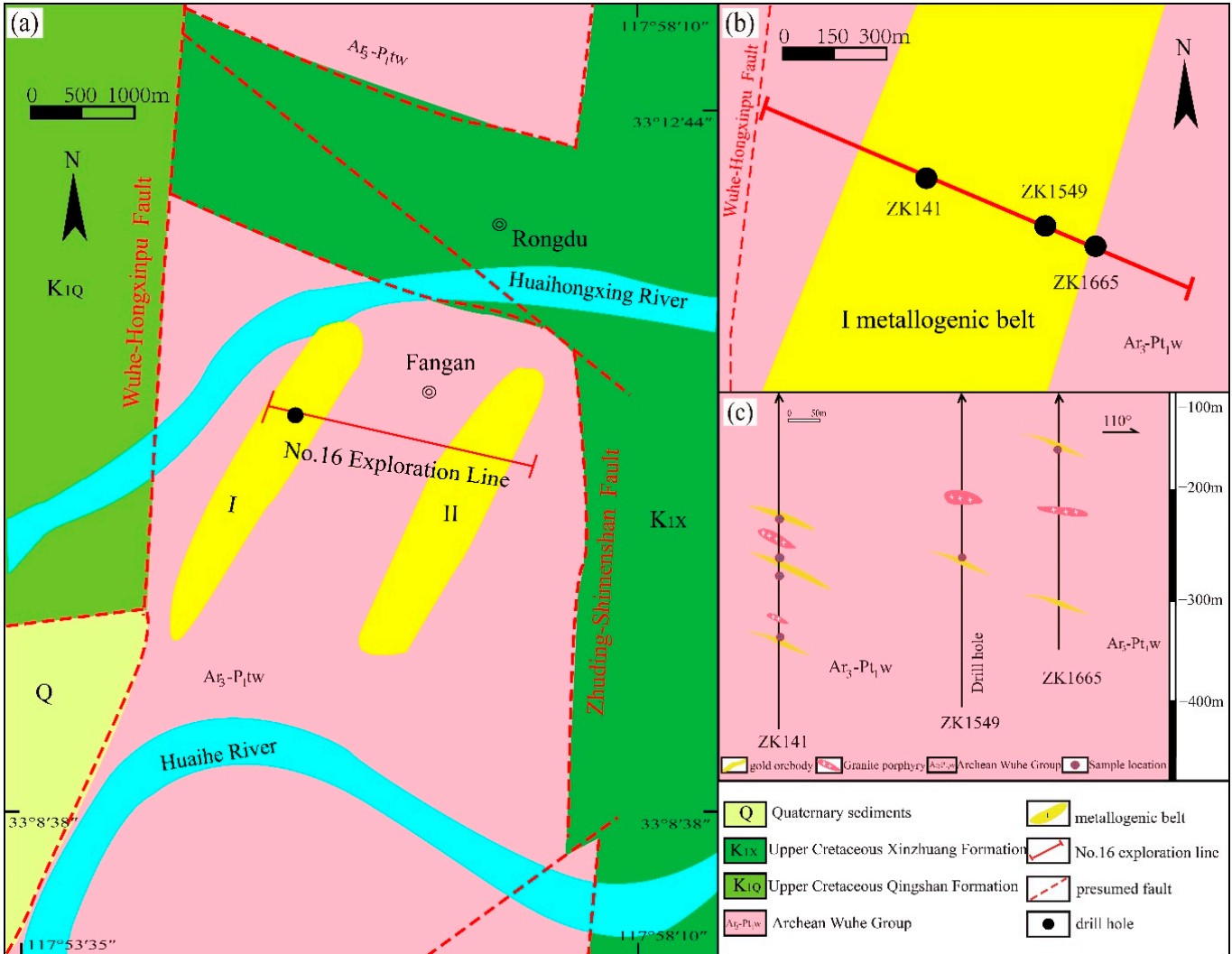

**Figure 2.** (**a**) Map of metallogenic belt distribution of Fang'an gold deposit, modified after [16]; (**b**) location of sampled drill holes projected onto the I metallogenic belt; (**c**) cross-section of No.16 exploration line in the Fang'an gold deposit, modified after [16] and No.312 Unit of Bureau of Geology and Mineral Exploration of Anhui Province.

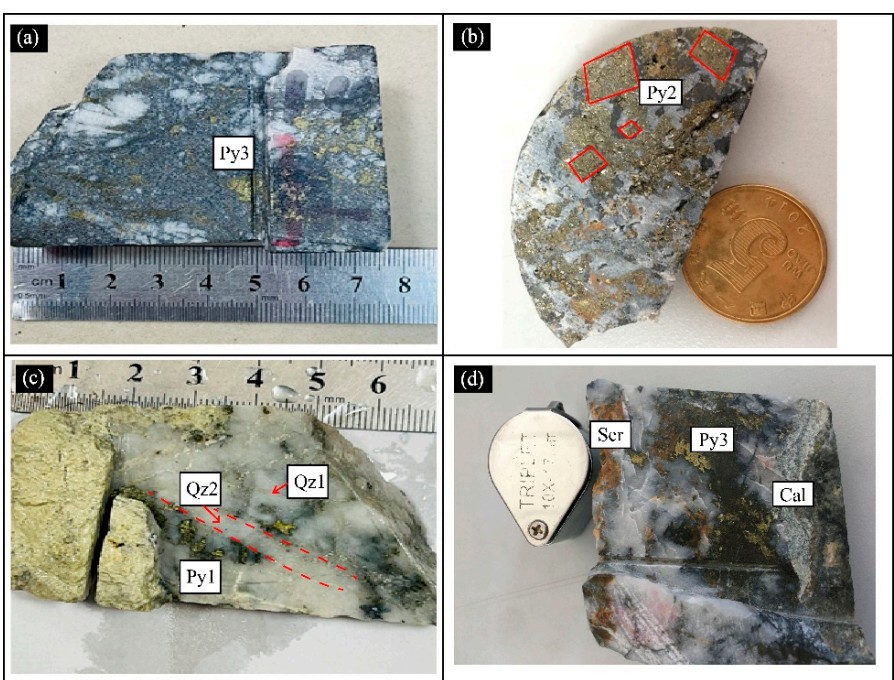

**Figure 3.** Ore features of Fang'an gold deposit: (**a**) polymetallic sulfide ore; (**b**) pyritohedron structure pyrite; (**c**) pyrite in quartz vein; (**d**) carbonate, quartz, and muscovite alteration in pyrite-quartz altered rocks. Py—pyrite; Qz—quartz; Ser—sercite; Cal—calcite.

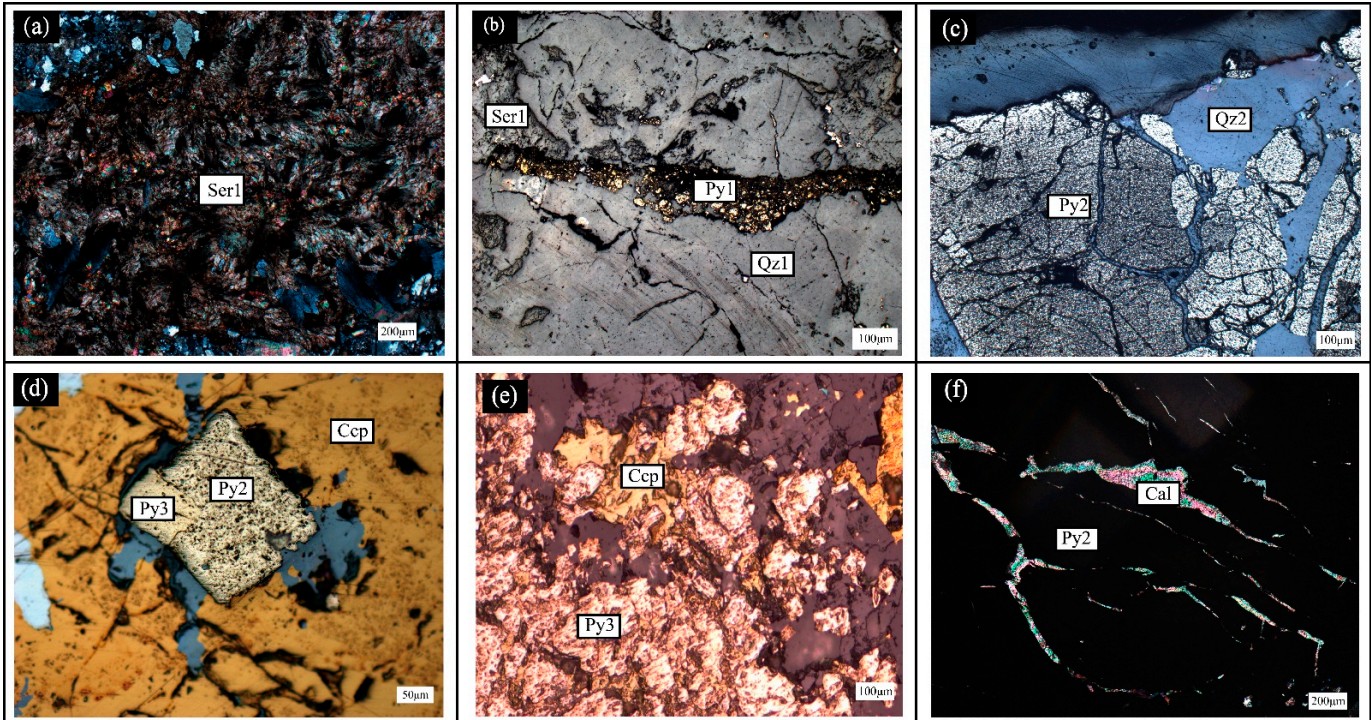

**Figure 4.** Ore microscope features of Fang'an gold deposit: (**a**) sericitization; (**b**) pyrites occurring in vein-filling form; (**c**) stockwork vein fracture in massive pyrite; (**d**) two-stage pyrite; (**e**) intergrown chalcopyrite in pyrite; (**f**) carbonation. Ccp—chalcopyrite; Py—pyrite; Qz—quartz; Ser—sercite; Cal—calcite.

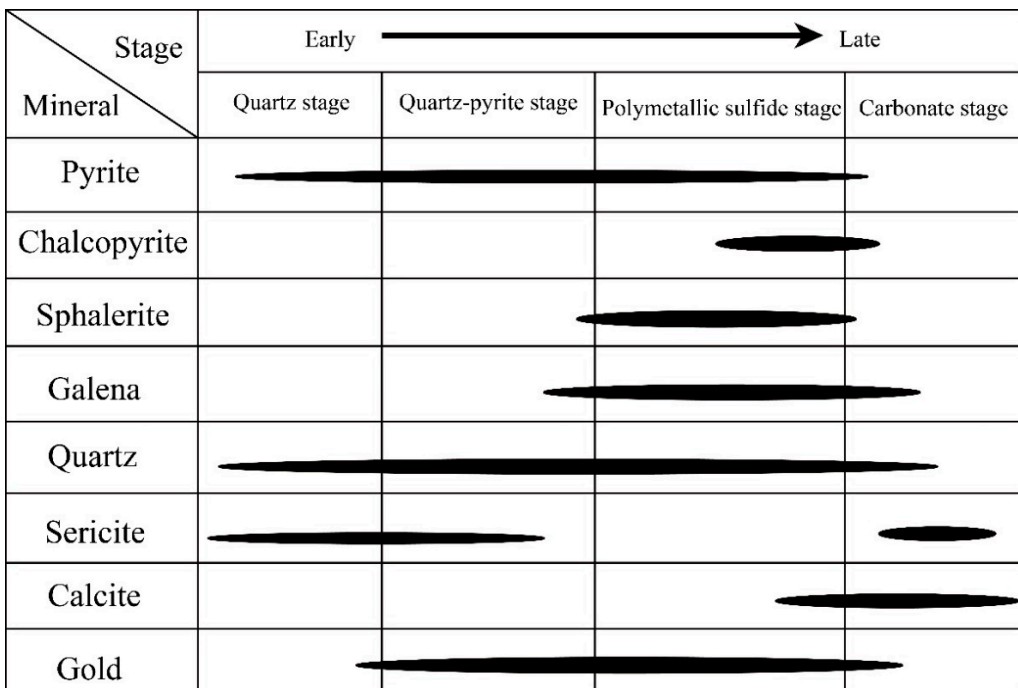

**Figure 5.** Mineral paragenetic sequence of the Fang'an gold deposit based on petrographic and BSE observations, as well as LA-ICPMS mapping.

## 3. Sampling and Analytical Methods

The samples were collected from Fang'an gold deposit No. I and mainly consisted of the ores and surrounding rocks in the drill holes ZK141, ZK1549, and ZK1665, which are widely distributed and representative. Based on the observation of the drill cores, seven ore samples were selected for the EPMA, and two samples were selected for LA-ICP-MS mapping.

The EPMA was used to analyze the composition of the pyrite with high efficiency and high resolution, and it can analyze a variety of elements with low loss, no pollution, and low interference as further advantages. It is widely used in mineralogical, petrological, and deposit research [25]. The analytical methods are described in detail below and briefly cited here [26]. Firstly, we made grinding microprobe slices and observed them by microscopy to find the classification and distribution of the minerals and elements. Then, we circled the minerals to be studied. The last step was to conduct carbon conduction treatment on the surface of the microprobe slices. The chemical composition of the minerals was analyzed by EPMA laboratory, School of Resources and Environmental Engineering, Hefei University of Technology. The instrument model was JOEL JXA 8230, and the analysis conditions were 15 kV (acceleration voltage), 20 nA (probe current), and 3~5 μm (spot size). The detection limits of the analytical elements Au, Ag, Cu, Fe, Pb, Zn, Cr, Co, Ni, Ga, and Ge were better than 0.1 wt%. The peak position and background counting time of the analyzed elements were 15 s and 5 s, respectively. Matrix corrections were performed by ZAF procedures.

LA-ICP-MS mapping was completed at the In-situ Mineral Geochemistry Laboratory, Ore Deposit and Exploration Centre, Hefei University of Technology, China. The analyses were carried out on an Agilent 7900 Quadrupole ICP-MS coupled to a Photon Machines Analyte HE 193 nm ArF Excimer Laser Ablation System.

## 4. Analysis Results

In this paper, the EPMA of gold-bearing pyrite from the Fang'an gold deposit was carried out by microscopic observation and BSE analysis, and a total of 47 points were measured. The results of the EPMA data for the different stages are presented in Table 1. The LA-ICP-MS mapping of the pyrite for the different stages is presented in Figure 6.

**Table 1.** Microprobe compositions of pyrites from Fang'an gold deposit.

| Stage | Point | Ga | Ge | S | Ag | Au | Ni | Co | As | Fe | Cu | Zn | Total |
|---|---|---|---|---|---|---|---|---|---|---|---|---|---|
| 1 | F4-4-PY-1 | 0 | 0.0281 | 49.4148 | 0.0041 | 0.0625 | 0.0754 | 0.0761 | 0 | 49.1491 | 0.1022 | 0.0069 | 99.6488 |
| 1 | F4-4-PY-2 | 0 | 0 | 49.8711 | 0.0189 | 0 | 0.0036 | 0.3949 | 0 | 48.8738 | 0.0149 | 0 | 99.9457 |
| 1 | ZK141-H11 | 0 | 0 | 48.158 | 0.028 | 0 | 0.025 | 0.091 | 0.049 | 50.142 | 0.054 | 0 | 99.128 |
| 1 | ZK141-H11 | 0 | 0 | 52.469 | 0.029 | 0.009 | 0 | 0.002 | 0.061 | 46.959 | 0 | 0 | 100.14 |
| 1 | N2-PY-1 | 0 | 0 | 51.2155 | 0.0118 | 0.0069 | 0.009 | 0.1385 | 0 | 47.2491 | 0.0177 | 0 | 99.3704 |
| 2 | F4-2-PY-5 | 0 | 0.0018 | 49.1633 | 0.0077 | 0.0416 | 0.0179 | 0.0159 | 0 | 49.1354 | 0 | 0.0157 | 99.1843 |
| 2 | F4-1-PY-1 | 0 | 0 | 50.1432 | 0 | 0.0832 | 0 | 0.039 | 0.005 | 48.297 | 0.1367 | 0.0088 | 99.506 |
| 2 | F4-1-PY-2 | 0 | 0.0053 | 49.4552 | 0.001 | 0.0902 | 0.0072 | 0.0142 | 0 | 48.9403 | 0.0121 | 0.0216 | 99.2505 |
| 2 | F4-1-PY-4 | 0 | 0 | 46.3357 | 0 | 0.0765 | 0.0394 | 0.0618 | 0 | 48.4876 | 0 | 0 | 95.7775 |
| 2 | ZK141-H13 | 0 | 0 | 52.46 | 0.027 | 0.049 | 0.041 | 0.098 | 0.111 | 46.347 | 0 | 0 | 96.866 |
| 2 | ZK141-H13 | 0 | 0 | 52.203 | 0.008 | 0.056 | 0 | 0.087 | 0.022 | 47.183 | 0.019 | 0 | 100.288 |
| 2 | H17-PY-3 | 0 | 0.0311 | 52.1037 | 0.0144 | 0 | 0.1528 | 0.1048 | 0 | 46.7288 | 0.0568 | 0.0088 | 100.0108 |
| 2 | ZK141-H13 | 0 | 0 | 52.051 | 0.042 | 0 | 0.034 | 0.044 | 0 | 47.189 | 0 | 0.057 | 100.046 |
| 2 | H17-PY-1 | 0.0371 | 0.0417 | 50.4985 | 0 | 0.1522 | 0.0126 | 0.0585 | 0 | 46.7403 | 0 | 0.0775 | 98.3235 |
| 2 | F4-2-PY-6 | 0.005 | 0 | 49.4862 | 0 | 0 | 0.0108 | 0.0584 | 0 | 49.0708 | 0.0177 | 0.0441 | 99.4674 |
| 2 | F4-2-PY-1 | 0.01 | 0 | 50.3203 | 0 | 0 | 0.1221 | 0.0567 | 0.0237 | 49.1287 | 0.0735 | 0 | 100.4499 |
| 2 | ZK141-H13 | 0.007 | 0 | 52.217 | 0.023 | 0 | 0 | 0.05 | 0.057 | 47.016 | 0.045 | 0.054 | 100.158 |
| 2 | ZK141-H13 | 0 | 0 | 52.287 | 0 | 0.013 | 0 | 0.073 | 0.064 | 47.294 | 0 | 0 | 100.429 |
| 2 | ZK141-H13 | 0 | 0 | 51.988 | 0 | 0.011 | 0 | 0.016 | 0 | 47.423 | 0.001 | 0.07 | 100.174 |
| 2 | ZK141-H13 | 0.017 | 0.014 | 51.694 | 0 | 0 | 0 | 0.085 | 0.051 | 47.369 | 0.05 | 0 | 99.957 |
| 2 | ZK141-H13 | 0.012 | 0 | 51.966 | 0 | 0 | 0 | 0.044 | 0.108 | 47.176 | 0.027 | 0 | 99.995 |
| 2 | ZK141-H13 | 0 | 0 | 52.841 | 0.029 | 0 | 0 | 0.046 | 0.046 | 46.076 | 0.024 | 0 | 99.711 |
| 2 | ZK141-H13 | 0.005 | 0 | 52.144 | 0 | 0 | 0 | 0.011 | 0.014 | 46.856 | 0 | 0.051 | 99.661 |
| 2 | ZK141-H13 | 0 | 0.01 | 52.333 | 0.021 | 0.041 | 0.056 | 0.02 | 0 | 46.235 | 0.056 | 0 | 99.415 |
| 2 | ZK141-H17 | 0 | 0.038 | 53.224 | 0 | 0 | 0 | 0.025 | 0 | 45.36 | 0.271 | 0.025 | 99.601 |
| 2 | F4-2-PY-7 | 0 | 0.007 | 50.2351 | 0 | 0 | 0 | 0.0656 | 0 | 48.9749 | 0.0325 | 0 | 100.0207 |
| 2 | F4-2-PY-3 | 0 | 0 | 50.2032 | 0 | 0 | 0.0862 | 0.0956 | 0.0174 | 48.9903 | 0.0009 | 0.0441 | 100.1669 |
| 2 | F4-2-PY-4 | 0 | 0 | 49.342 | 0 | 0 | 0.0072 | 0.0106 | 0.203 | 48.8755 | 0.0446 | 0 | 99.2679 |



**Table 1.** *Cont.*

| Stage | Point | Ga | Ge | S | Ag | Au | Ni | Co | As | Fe | Cu | Zn | Total |
|-------|-------|----|----|---|----|----|----|----|----|----|----|----|-------|
| 3 | 1549-PY-2 | 0 | 0 | 52.33 | 0.0201 | 0.0415 | 0 | 0.0462 | 0 | 47.0166 | 0.0121 | 0 | 100.2038 |
| 3 | ZK141-H6 | 0 | 0 | 50.451 | 0 | 0.019 | 0 | 0.069 | 0 | 48.152 | 0.123 | 0 | 99.513 |
| 3 | ZK141-H6 | 0 | 0 | 51.615 | 0.015 | 0.021 | 0 | 0.082 | 0.016 | 47.427 | 0.034 | 0 | 99.975 |
| 3 | ZK141-H6 | 0 | 0.01 | 52.506 | 0.029 | 0 | 0 | 0.06 | 0.009 | 46.61 | 0.337 | 0 | 100.246 |
| 3 | ZK141-H1 | 0 | 0.05 | 51.433 | 0 | 0.026 | 0 | 0.053 | 0 | 46.689 | 0.003 | 0.069 | 99.069 |
| 3 | ZK141-H4 | 0 | 0 | 53.031 | 0.019 | 0.06 | 0 | 0.042 | 0.007 | 46.156 | 0 | 0.046 | 100.059 |
| 3 | ZK141-H11 | 0.02 | 0.034 | 52.731 | 0.019 | 0 | 0.046 | 0.111 | 0 | 46.675 | 0.056 | 0 | 100.372 |
| 3 | ZK141-H11 | 0.012 | 0.033 | 51.712 | 0.042 | 0 | 0.021 | 0.037 | 0 | 47.762 | 0.056 | 0 | 100.326 |
| 3 | H17-PY-2 | 0 | 0 | 52.5273 | 0 | 0.0345 | 0 | 0.0444 | 0.0442 | 46.5909 | 0.0456 | 0.0619 | 100.087 |
| 3 | H17-PY-4 | 0 | 0 | 52.0677 | 0.0397 | 0.0345 | 0.0503 | 0.0994 | 0.0369 | 46.34 | 0.1396 | 0 | 99.7345 |
| 3 | F4-1-2-PY-2 | 0 | 0 | 50.3255 | 0.0072 | 0 | 0.0287 | 0.0514 | 0.0199 | 48.712 | 0.0474 | 0.0088 | 99.9712 |
| 3 | F4-1-2-PY-3 | 0 | 0.0228 | 49.6464 | 0.0066 | 0.0347 | 0.0789 | 0.0973 | 0.0623 | 48.6983 | 0.053 | 0.0441 | 99.4791 |
| 3 | F4-1-2-PY-4 | 0 | 0.0508 | 49.1736 | 0 | 0.0069 | 0.0269 | 0.023 | 0 | 48.3353 | 0.0474 | 0 | 98.4057 |
| 3 | F4-1-3-PY-2 | 0 | 0 | 49.2931 | 0.0097 | 0 | 0 | 0.092 | 0.0486 | 48.5112 | 0.0567 | 0 | 98.7843 |
| 3 | F4-1-3-PY-3 | 0 | 0 | 49.3239 | 0 | 0 | 0.0251 | 0.0673 | 0 | 48.6918 | 0.0688 | 0.0049 | 98.9047 |
| 3 | F4-1-3-PY-4 | 0 | 0.0123 | 49.2288 | 0 | 0 | 0 | 0.0584 | 0.005 | 48.6244 | 0.0762 | 0.0265 | 98.745 |
| 3 | 1549-PY-1 | 0.0222 | 0 | 52.3566 | 0.0015 | 0.0553 | 0.0396 | 0.048 | 0 | 47.1191 | 0.0475 | 0.0314 | 100.4698 |
| 3 | F4-1-2-PY-6 | 0 | 0 | 49.9961 | 0.0312 | 0 | 0.6241 | 0.8695 | 0.0062 | 47.2792 | 0.0753 | 0 | 99.6443 |
| 3 | 1549-PY-3 | 0 | 0 | 52.7541 | 0.0015 | 0 | 0.0395 | 0.0888 | 0.1303 | 46.8642 | 0.0978 | 0.0913 | 100.8182 |

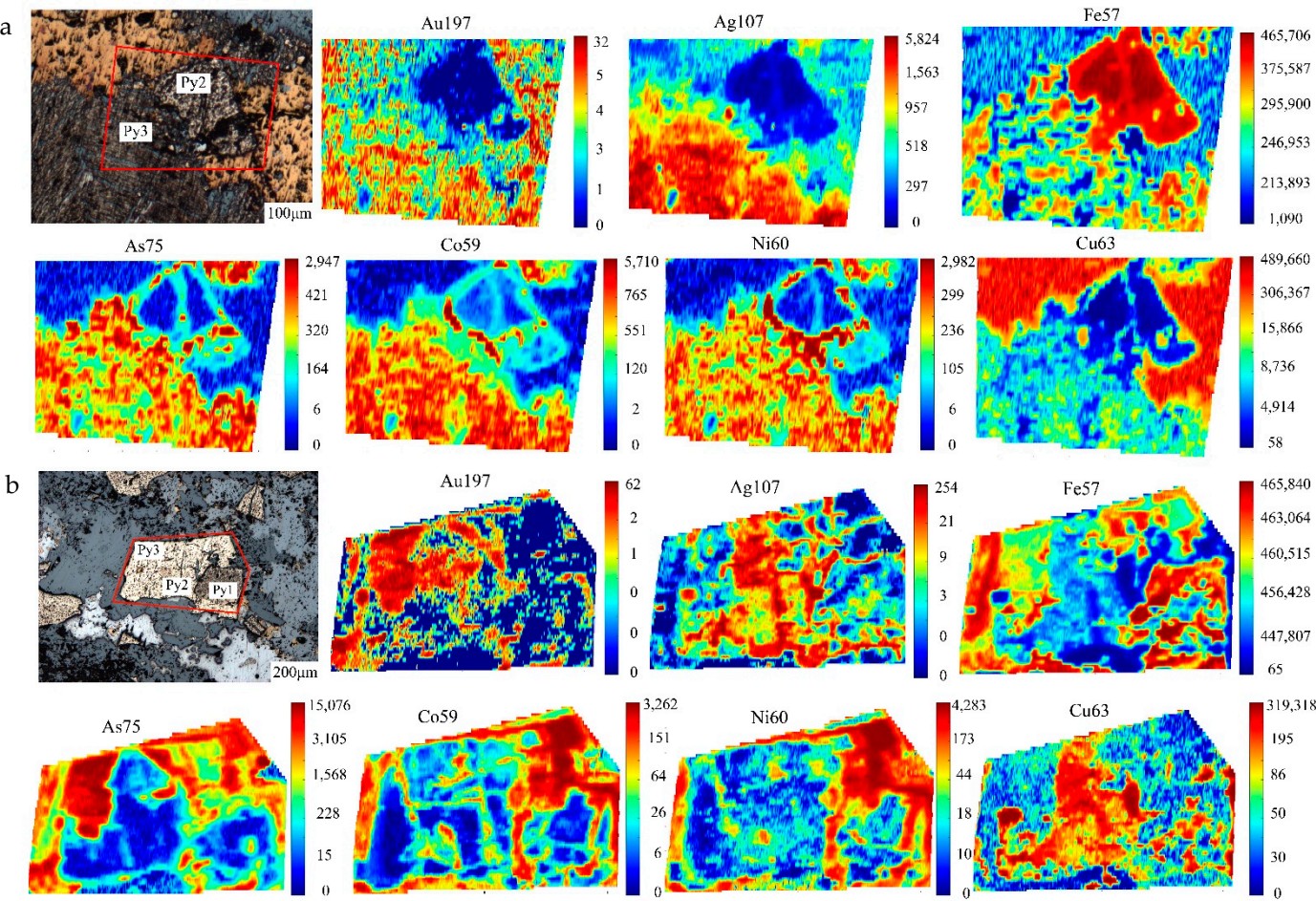

**Figure 6.** Laser ablation inductively coupled plasma mass spectrometry mapping of trace elements (Au197, Ag107, Fe57, As75, Co59, Ni60, and Cu63) in pyrite of Fang'an deposit: (**a**) showing trace element contents of Py2 and Py3 in sample ZK141-H1; (**b**) showing trace element contents of Py1, Py2, and Py3 in sample ZK1549.

For pyrite in the quartz stage (I), w(Fe) varies within a range of 46.95%~50.14%, with an average value of 48.47%, which is higher than its theoretical value of 46.55%. The w(S) varies within a range of 48.15%~52.46%, with an average value of 50.22%. The w(As) varies within a range of 0%~0.061%, with an average value of 0.055%. In this stage, the trace elements of pyrite were Cu, Au, and Ag, and their values were within 0%~0.102%, 0%~0.062%, and 0.004%~0.029%, respectively. Furthermore, pyrite in this stage contains a small amount of Co and Ni, with w(Co) and w(Ni) within the ranges of 0.002%~0.395% and 0%~0.075%, respectively. The Co/Ni ratios were higher than 1, and some of the data varied widely.

In the quartz-pyrite stage (II), the w(Fe) was within a range of 45.36%~49.16%, with an average value of 47.6%, which is higher than its theoretical value of 46.55%. The w(S) varied within a range of 46.34%~53.22%, with an average value of 51.07%. The w(As) varied within a range of 0%~0.203%, with an average value of 0.06%. In the quartz-pyrite stage, the trace elements of pyrite were weak, and the Cu, Au, and Ag values were within 0%~0.271%, 0%~0.152%, and 0.004%~0.042%, respectively. Furthermore, the w(Co) and w(Ni) were within the ranges of 0.011%~0.105% and 0%~0.153%, respectively. The Co/Ni ratios were 0.357~5.407, with an average value of 1.942.

In the polymetallic sulfide stage (III), the w(Fe), w(S), and w(As) were within the ranges of 46.34%~48.70%, 49.17%~52.75, and 0%~0.13%, respectively, with average values of 47.49%, 51.18%, and 0.035%, respectively. In the polymetallic sulfide stage, the trace

elements of pyrite were Cu, Au, and Ag and the values were within 0%~0.14%, 0%~0.06%, and 0%~0.042%, respectively, with average values of 0.078%, 0.033%, and 0.018%, respectively. Moreover, the w(Co) and w(Ni) were 0.002%~0.11% and 0%~0.62%, respectively. The Co/Ni ratios were 0.855~2.68, and the average value was 1.756.

The LA-ICP-MS mapping of pyrite in the three stages (Figure 6) shows internal heterogeneity with respect to the trace elements. The heterogeneity reflects compositional zoning in terms of the trace elements; as well as Py1, Py2, and Py3, the various pyrite generations display overgrowth textures where pyrite from stage 2 (Py2) rims the brecciated Py1. Py3 is also observed to rim Py2 (Figure 6b), which is common in ore adjacent to composite veins.

The LA-ICP-MS mapping shows that multiple elemental (Au197, Ag107, Fe57, As75, Co59, Ni60, and Cu63) distributions across the individual pyrites occur as intensive zones (Figure 6). Py1 has low concentrations of As and other trace elements (e.g., Co, Ni). Py2 contains the highest Cu and relatively high Au and Ag (Figure 6a). Py3 is marked by high Au and As and relatively high Co and Ni and hosts some submicron-sized particles of native gold (Au°) and sulfides of Cu. The mapping shows a gradual increase in gold content from Py1 to Py3, with the highest gold content in Py3 corresponding to the polymetallic sulfide stage of the mineralization.

## 5. Discussion

The gold-bearing minerals of quartz-vein-type gold deposits are mainly pyrite and quartz, and pyrite formed under different physical–chemical conditions has differences in its crystal morphology and trace element content [9,27,28]. The type, content, elemental association, and elemental ratio of the major and trace elements in pyrite are important typomorphic characteristics. The trace elements in pyrite are captured in the formation process, and the content is closely related to the ore fluid and the physicochemical conditions [27]. Therefore, through EPMA analysis, the composition changes during the formation of pyrite can be revealed, and the contents of major trace elements in pyrite can be determined by calculating the temperature–pressure conditions, indicating the genetic type of the deposit [29,30].

Pyrite is one of the important gold-bearing minerals [31–33]. The geochemistry of gold has strong chalcophile and siderophile qualities, and gold tends to be enriched in pyrite. Additionally, Au elements migrate in the form of complexes. When sulfide crystals such as arsenopyrite and pyrite grow, the concentration of S around it decreases; then, the Au-S complexes are easy to decompose, leading to the gold being attached to the crystal growth surface. Therefore, with pyrite crystallization, gold can easily enter the crystal inside; so, the gold content is high. Due to this, pyrite is the main gold-bearing mineral of gold deposits.

According to previous studies, as the major elements in pyrite, the S and Fe content and S/Fe ratio are closely related to ore genesis [34]. The values of the trace elements of Co/Ni indicate the ore genesis and mineralization [33,35,36], and the Au/Ag ratio reflects the type of ore genesis. Reich et al. (2005) studied several gold deposits in the United States and found that the contents of As and Au indicate the occurrence state of gold [37]. Li et al. (2005) concluded that a higher Au content in pyrite occurs in magmatic–hydrothermal-type gold deposits with Au/Ag $\geq$ 0.5, such as the volcanic type, the structural–alteration type, the sedimentary–metamorphic–hydrothermal–replacement type, and various other types of associated gold deposits with Au/Ag < 0.5 [32]. Deditius et al. (2014) reported that there is a positive correlation between Au and As because As replaces S in pyrite, resulting in pyrite lattice distortion and Au$^+$ in the pyrite lattice [38]. The cupreous ions of Cu$^{2+}$, Pb$^{2+}$, and Zn$^{2+}$ are different from the Fe$^{2+}$ ions, which cannot easily enter into the pyrite lattice through homomorphism. If the pyrite is rich in cupreous ions such as Cu$^{2+}$, Pb$^{2+}$, and Zn$^{2+}$, indicating that Pb, Zn, and Cu are mainly present as sulfide inclusions, there is a nonlinear relationship between Au and the cupreous elements [32]. In summary, the contents and

correlations of major and trace elements in pyrite can reflect the physicochemical conditions and metallogenic backgrounds.

In hydrothermal systems, polymetallic melts can efficiently capture metals from hydrothermal fluids, representing a highly efficient way to enrich low-abundance ore components, such as gold [39]. However, when the host mineral continues growing, the polymetallic melts formed through the adsorption–reduction mechanisms on the mineral surface may become trapped in the host mineral and thus isolated from the fluid. Polymetallic melts could remain molten at very low temperatures and, if able to migrate, contact with the fluids again. As a result, polymetallic melts can repeatedly act as gold collectors in hydrothermal gold deposits, which are recorded as hydrothermal events or Au mineralization [40–42]. Gourcerol (2018) reported that the Au versus Ag plot is used to track Au mineralization and hydrothermal events [43]. The hydrothermal event has up to 100 ppm Au present in the sulfides, likely as invisible Au or nanoparticles [44]. The subsequent hydrothermal events are recorded by an upgrading due to the zone refining of the early Au, with it either having a similar Au/Ag ratio or a much higher one.

The correlation analysis results of the trace elements of pyrite in the Fang'an gold deposit showed that Au and Ag are positively correlated (Figure 7). Regarding the Au and Ag concentrations in pyrite, we note the Py3 surface with many fractures and containing micro-inclusions, as demonstrated by the LA-ICP-MS mapping observations, and higher concentrations of these elements than in the Py2, as is clearly evident in Figure 6a. Thus, the element maps, which offer complementary information, provide evidence for a polymetallic sulfide stage enrichment of Au and Ag. In the studied samples, the LA-ICP-MS mapping of Au and Ag in the pyrite (Figure 6a) suggests that nanoparticles of Au and Ag are in existence. Meanwhile, high concentrations of Au (69~1522 ppm) and Ag (10~420 ppm) can be detected by EPMA, indicating that major lattice-bound Au and Ag occur in the pyrite. Based on the element-distribution mapping, the Au shows a similar distribution to that of Cu (Figure 6), indicating that a certain amount of Au is associated with the metal sulfides (e.g., chalcopyrite). These metal sulfides have been widely identified in the primary ores of the Fang'an gold deposit. This polymetallic sulfide stage of precious-metal enrichment relates to the post-crystallization stage of the pyrite grains and reflects a potential upgrading of Au and, to a lesser extent, Ag due to either continued and or subsequent fluid circulation [45–47].

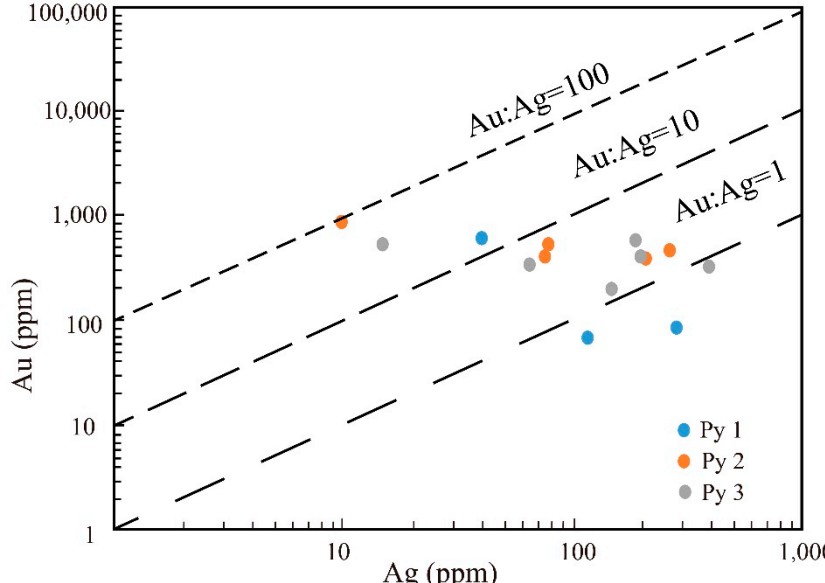

**Figure 7.** Binary element plots of Au versus Ag plots (in ppm) of pyrite grains in Fang'an gold deposit.

*5.1. Characteristics of Fe and S in Pyrite*

Through studying the major element typomorphic characteristics of pyrite, using the S/Fe ratio to indicate the mineralization, the pyrite could be divided into two types: enriched in Fe and depleted in S (S/Fe < 2) and enriched in S and depleted in Fe (S/Fe > 2) [32]. According to the previous statistical analysis of the composition of pyrite in gold deposits of different genesis types, volcanic–hydrothermal, sub-volcanic–hydrothermal, magmatic–hydrothermal, and Carlin-type are rich in Fe and poor in S (S/Fe < 2), and the average content chemical formula is $FeS_{1.97–1.99}$. The metamorphic hydrothermal type is rich in Fe and S, and the average content chemical formula is $FeS_{1.95}$. The sedimentary genetic-type (S/Fe > 2) pyrite formula is $FeS_{2.03}$ [33,36,48,49].

Chen et al. (1989) studied the Linglong deposit in the Jiaodong area and found that the S/Fe was within a range of 1.91~2.18, with an average value of 1.95 [27]. The pyrite composition was enriched in Fe and depleted in S. S loss is the result of $As^{3-}$, $Sb^{3-}$, and $S^{2-}$ occurring in homomorphism in pyrite, which makes the pyrite crystal structure appear vacant, thus increasing the structural defects and allowing the gold to enter the pyrite. Therefore, the loss of S is beneficial to the enrichment of gold in pyrite [32].

In the Fang'an gold deposit, the w(Fe) varies within a range of 45.36%~50.142%, with an average value of 47.65%, which is higher than its theoretical value of 46.55%. The w(S) is within a range of 46.336%~53.224%, with an average value of 51.03%, which is lower than its theoretical value of 53.45%. The atomic ratio value is within the range of 1.664~2.043, with an average value of 1.867, S/Fe < 2. This means that the pyrite is enriched in Fe and depleted in S in the Fang'an gold deposit, indicating that the genesis is related to the volcanic–hydrothermal type, the sub–volcanic–hydrothermal type, the magmatic–hydrothermal type, and the Carlin type. The quartz D–O isotopes in other gold deposits in the Bengbu uplift ranged from −58.7 to −90‰ (δD) and 11.6 to 14.6‰ ($\delta^{18}O$), suggesting that the ore-forming fluids were dominated by magmatic water, which could relate to the mantle source [4,10]. Previous researchers used He–Ar isotopes from early Cretaceous gold deposits in the NCP to show that the ore-forming fluids were predominantly derived from the mantle [5,28]. Therefore, we speculate that the hydrothermal fluids of the Fang'an gold deposit were mainly exsolved from mantle-derived magmas.

*5.2. Characteristics of Co and Ni in Pyrite*

Usually, the contents of Co and Ni are related to the physicochemical conditions of pyrite formation. The study of the geochemical characteristics of w(Co), w(Ni), and the Co/Ni ratio in pyrite is beneficial for distinguishing its genesis as magmatic, hydrothermal, or sedimentary [27,48].

Ni is a siderophile element, and its precipitation rate is less than that of Fe, which is more likely to enter the pyrite lattice. In reduction environments, Ni is less active and is released when the crystals recrystallize [50–52]; so, the content of Ni in pyrite can provide metallogenic materials information. Palme H (1996) calculated that the Ni content is about $(2200 \pm 500) \times 10^{-6}$, according to mantle rocks [53]. The Ni content of felsic rocks, such as highly differentiated magmatic rocks (granites), is usually very low, and the Ni content of the continental crust ranges from $19 \times 10^{-6}$ to $60 \times 10^{-6}$ [54]. The Ni content in pyrite associated with late granitic hydrothermal fluids is generally very low. In this paper, the Ni content of the gold-bearing pyrite in the Fang'an gold deposit was analyzed by the EPMA as being between $18 \times 10^{-6}$ and $1528 \times 10^{-6}$ (Table 1), which is in the range of mantle rocks and the continental lower crust [54]. This is consistent with the findings of Hu et al. (2015) and Zhang et al. (2017), who suggested that the mineralized material of gold deposits in the Wuhe area has been derived from the lower crust [11,14].

Xu et al. (1980) analyzed and compared the Co/Ni of pyrite in domestic metal sulfide deposits and concluded that Co/Ni < 1 has a sedimentary genesis [48], Co/Ni > 1 is volcanic and magmatic–hydrothermal, and metamorphism will increase and disperse the Co/Ni values in pyrite. Therefore, the Co/Ni ratio is an effective indicator for distinguishing the genesis of different deposits.

In the Fang'an gold deposit, the pyrite w(Co)/w(Ni) ranged from 0.357 to 109.69. Among them, the four measured points of the quartz-pyrite stage (Py1) had Co and Ni levels that were higher than the detection limit, and all four points had Co/Ni values greater than 1, with hydrothermal modification characteristics. Two of them had large data differences greater than 10, which indicates metamorphism. The Co/Ni ratio of the quartz-pyrite stage pyrite (Py2) ranged from 0.357 to 5.407, with a mean value of 1.942. Four of the points with w(Co)/w(Ni) of less than 1 indicated that the native composition characteristics were maintained, while the remaining eight points with Co/Ni greater than 1 had hydrothermal modification characteristics. In the polymetallic sulfide stage (Py3), there were ten measured points with Co and Ni above the detection limit, and the Co/Ni ratio ranged from 0.855 to 2.68, with an average value of 1.756. Only one point of w(Co)/w(Ni) was less than 1, while the remaining nine points of Co/Ni were all greater than 1. Pyrite in this stage was the same as the hydrothermal fluid in the quartz-pyrite stage. It was confirmed that the pyrite was formed by hydrothermal action, with the participation of a small amount of magmatic material. Projecting the Co/Ni values of pyrite into the Co/Ni diagram (Figure 8), most of the samples fell in the range of hydrothermal genesis and some fell in the range of volcano genesis, indicating that the gold-bearing pyrite in the Fang'an gold deposit is mainly influenced by hydrothermal action. Some of the data of the volcanic and sedimentary genesis may be caused by the original rocks of the enclosing Xigudui formation, containing certain submarine volcanic eruption materials during deposition. Therefore, the genesis of the Fang'an gold deposit is of the hydrothermal type, and its material composition is affected by the surrounding rock to a certain extent. The alteration types of the Fang'an gold deposit are pyritization, silicification, sericitization, and carbonation, showing the medium-low temperature hydrothermal alteration combination.

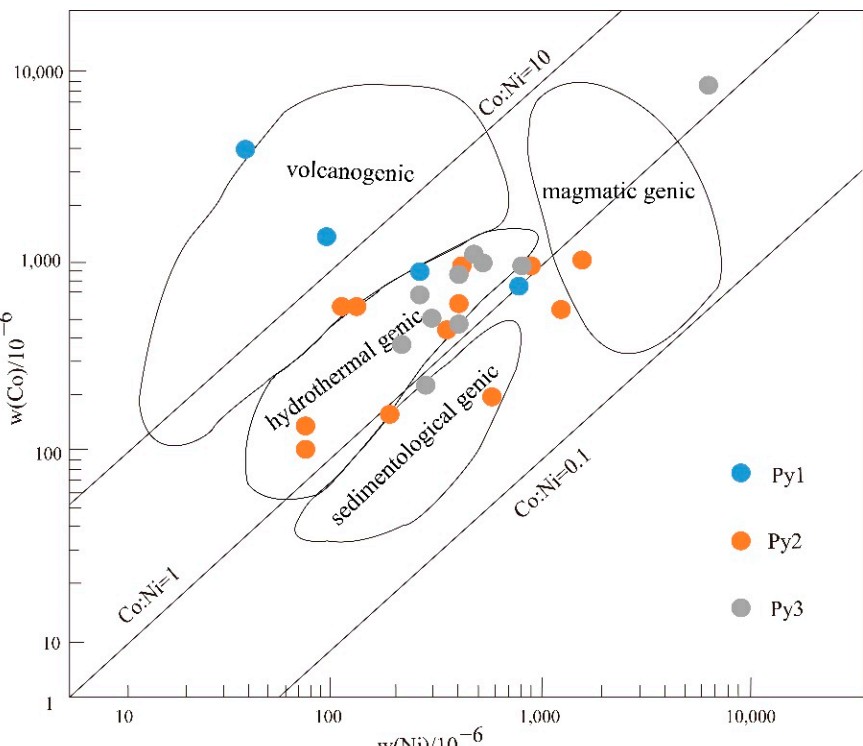

**Figure 8.** Co/Ni diagram from gold-bearing pyrite (atomic percentage, fitting equation according to [49]).

*5.3. Gold Occurrence*

Usually, As is closely related to Au [55]. Reich et al. studied several As-bearing gold deposits in the United States and proposed that invisible gold has two forms, nanometer-sized particles of gold (Au°) and gold solid solution (Au⁺), and the practical calculation for-

mulas for the solubility curve of Au in pyrite were established ($C_{Au} = 0.02C_{As} + 4 \times 10^{-5}$) (Figure 9) [37]. The gold in As-bearing pyrite is located in the upper region of the solubility curve (Au/As > 0.02), and its major occurrence is nanometer-sized particles of gold (Au°). Located in the lower region of the solubility curve (Au/As < 0.02), the major occurrence is the gold solid solution (Au$^+$). Zhao et al. (2020) researched the Shuiyindong gold deposit and Taipingdong gold deposit. The correlation between As and Au showed that As is not always positively correlated with Au. It is not conducive to Au enrichment when the As content is too high or too low. A $W_{As}$ within a range of 0.05~0.1 is conducive to Au enrichment [56].

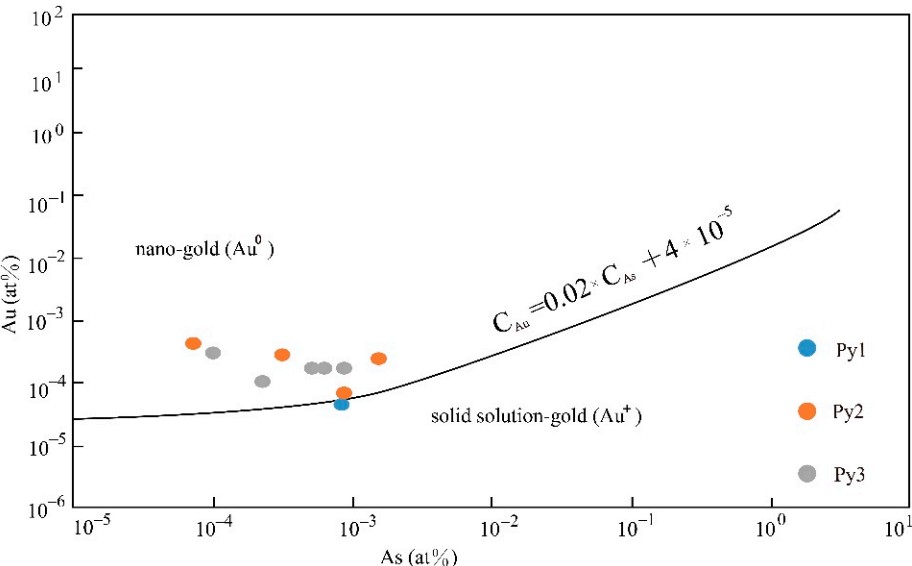

**Figure 9.** Relationship diagram of As and Au content in pyrite (atomic percentage, fitting equation according to [48]).

Studying the Fang'an gold deposit, native gold particles were not found by microscopy and BSE images, but the EPMA data showed that some measurement points contained gold, indicating that the gold was unevenly distributed at the resolution of the EPMA and implying gold enrichment in the form of invisible gold. In the quartz-pyrite and polymetallic sulfide stage, the $W_{As}$ average values were 0.06% and 0.018%, respectively, and most were between 0.01 and 0.1. It is conducive to Au enrichment when the $W_{As}$ values are within the range of 0.05~0.1. Using As-Au atomic percentage data to create images (Figure 9), we found that the measured points of the quartz-pyrite stage and the polymetallic sulfide stage fell in the nano-gold (Au°) region, while the measured points of the quartz stage were within the gold solid solution (Au$^+$) region, indicating that the quartz-pyrite stage and the polymetallic sulfide stage are enriched Au elements. Therefore, the gold occurrence is nanometer-sized particles of gold with uneven disseminated distribution in the Fang'an gold deposit.

### 5.4. Gold Mineralization

The correlation analysis results of trace elements of pyrite in the Fang'an gold deposit showed that the added trace elements have had no significant effect on the enrichment of gold in pyrite, indicating that the wall rocks have had little influence on mineralization. From the previously derived S/Fe < 2, we know that the pyrite in the Fang'an gold deposit is rich in Fe and poor in S. This indicates that the ore-forming fluids are hydrothermal. Meanwhile, the Co/Ni ratio is generally greater than 1, indicating that the ore-forming fluids are mainly derived from magmatic–hydrothermal fluid. This is consistent with previous H-O isotope experiments on quartz from the Rongdu gold deposit in the eastern Bengbu uplift, which found that the mineralizing fluid was derived from magmatic–hydrothermal

fluids [9]. Moreover, at the Nvshan intrusion, located 6 km south of the ore field, where the megaporphyritic biotite granite was formed at an age of 130.1 ± 3.2 Ma, the megaporphyritic biotite granite was formed in the magmatic event of the Early Cretaceous [18]. Yang et al. (2019) believe that the Re-Os isotope dating of pyrite from the Rongdu gold deposit shows that the mineralization age is 134 ± 19 Ma [8], which coincides with the age of the granite-type formation around 130 Ma in the area. Known gold deposits in this area usually have close relationships with these Yanshanian magmatic dikes in space. Thus, this paper suggests that the formation of the Fang'an gold deposit was associated with magmatic activity at about 130 Ma. In addition, the eastern NCP records the subduction event of the Paleo-Pacific plate during the Late Mesozoic causing the large-scale tectonic extension along the Tan-Lu fault [57]. The strong extensional movement induced partial melting of the lithospheric mantle and the lower crust, generating gold-related mafic (~130 Ma) and granitic melts (115–130 Ma) [5,8], respectively. In addition, the pre-existing NE–NNE striking faults adjacent to and inside of the Tan-Lu fault reactivated as normal faults in this stage [3]. These gentler dipping faults provided structural sites for precipitation of the gold in the magmatic–hydrothermal fluids. The mineralization process was due to the process of the upward transport of the magmatic–hydrothermal fluid. Au elements migrate in the form of complexes; so, when pyrite and other sulfide crystals grow, the surrounding sulfur concentration decreases, and the Au-S complexes can easily decompose so that gold is attached to the growth surface of the crystal. This makes the gold content of pyrites very high. In the process of upward transport, temperature reduction enables rapid cooling of the pyrite, resulting in a significant increase in structural defects, which makes it more suitable for the precipitation of Au. This is a phenomenon that results in an increase in Au content in the quartz-pyrite stage and the polymetallic sulfide stage.

## 6. Conclusions

The following conclusions were drawn from the pyrite in different mineralization stages of the Fang'an gold deposit reported in this study. The Fang'an gold deposit has four stages: the quartz stage, the quartz-pyrite stage, the polymetallic sulfide stage, and the carbonate stage. Py1, Py2, and Py3 formed in the first three stages, respectively. Then, we found by EMPA analysis that the pyrite generally contained Au in the first three mineralization stages, indicating that pyrite is one of the main gold-bearing minerals in the Fang'an gold deposit. The LA-ICP-MS mapping illustrates that gold gradually increases from Py1 to Py3 in the pyrite of phase III, with the highest gold content in Py3. It can be seen that gold is mainly endowed in the phase III polymetallic sulfide stage. The Fe/S ratio of pyrite showed that the Fang'an gold deposit is a hydrothermal quartz vein type. The general Co/Ni > 1 of pyrite suggests that the ore-forming fluids are magmatic–hydrothermal. Moreover, this paper suggests that the formation of the Fang'an gold deposit was associated with magmatic activity at about 130 Ma. The mineralization process was due to the process of the upward transport of magmatic–hydrothermal fluid. Au migrates in the form of complexes, making the gold content higher in pyrite, and the gold occurrence is nanometer-sized particles of gold (Au°).

**Author Contributions:** Conceptualization, Y.W. and S.R.; methodology, Y.W. and S.R.; formal analysis, Y.W., Z.Z., L.X., and J.W.; investigation, Y.Z.; data curation, J.W. and G.Z.; writing—original draft preparation, Y.W.; writing—review and editing, Y.W.; supervision, C.S.; funding acquisition, S.R. All authors have read and agreed to the published version of the manuscript.

**Funding:** This research was funded by the Public Welfare Project of Anhui Province (grant No. 2016-g-4).

**Informed Consent Statement:** Informed consent was obtained from all subjects involved in the study.

**Data Availability Statement:** The data that support the findings of this study are available from the corresponding author upon reasonable request.

**Conflicts of Interest:** The authors declare no conflict of interest.

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
