# Peer review of "Typomorphic Characteristics of Gold-Bearing Pyrite and Its Genetic Implications for the Fang’an Gold Deposit, the Bengbu Uplift, Eastern China"

_minerals, doi:10.3390/min12101196_

Round 1
Reviewer 1 Report
Some suggestions for improving the MS.
1. In the abstract, it's stated that the the source of ore-forming fluid of the Fang'an deposits, whereas, there is little contents on the ore-forming fluid and ore material in the Discussion. Some data and discussions shouled be added.
2.In the part of Introduction, Some viewpoints of ore genesis of gold deposits in this area should be added, which is important for understanding implications of this MS.
3.Besides the Fang’an, other gold deposits involved in this sections, Mao shan, Rongdu and Zhu Ding should be marked in Figure 1.
4.In L55, the Fang’an gold deposit is a medium-sized quartz-vein type gold deposit, however, in the L93, authors present the proven reserves of gold are 479kg, with an average grade of 2.63 g/t. Please check these data.
5. In Discussion, although some conclusions, including the Fang’an gold deposit was associated with magmatic activity at about 130Ma, the mineralization process was due to the process of upward transport of magmatic–hydrothermal fluid, some associated evidences of geology and deposit should be added.
6.The English of this MS has some unclear or inaccurate usage, the English need polished.
Other revision suggestions are marked in the MS-reviewer marked.

Reviewer 2 Report
The manuscript is undoubtedly of interest to those who are interested in the problems of gold metallogeny and the typomorphism of ore minerals in gold deposits. Therefore, the article can be recommended for publication in the journal, despite the locality of the material. The data presented in the article are original and make a certain contribution to the direction of studying the typomorphism of pyrite, as the main mineral of gold deposits. The authors quite clearly identified three stages of pyrite formation and showed features of its composition. To a lesser extent, this concerns the distribution of impurities, where more statistics and the use of a different method of study, such as ICP-MS or Laser microanalysis, are required. As a comment on the content of the article, I note that the "Discussion" section there is no explanation of the reasons for the difference in the composition of pyrite-1 from pyrites of two later generations. Pyrite-1, according to the authors, is maximally ferruginous and has a different Co-Ni ratio trend to pyrites-2 and 3 (lines 176-201, fig. 6). This section should be explain these differences. General arguments about the nature of pyrite are clearly not enough here. But in general, after taking into account my comments and wishes, the article can be accepted for publication.
Reviewer 3 Report
Manuscript ID: minerals-1842019
Type of manuscript: Article
Title: Typomorphic characteristics of gold-bearing pyrite and its genetic implications for the Fang’an gold deposit, the Bengbu Uplift, Eastern China.
Authors: Ying Wang, Sheng Lian Ren *, Ze Zhong, Li Xiong, Gang Zhang, Juan Wang, Yan Zhang, Chuan Zhong Song
The manuscript devoted to the study of geological characteristics, the major and trace elements of pyrite in different stages, to explore the compositional characteristics of pyrite, the occurrence of gold, and the source of ore-forming fluid. Authors found that the deposits can be divided into four ore-forming stages: the quartz stage, quartz-pyrite stage, polymetallic sulfide stage, and carbonate stage. The pyrites can be divided into three stages, corresponding to the first three ore-forming.
I therefore recommend it for publication subject to major revision summarize here and minor revisions see in attached PDF file:
1- It is not clear how informative this study is, since the work does not contain information about the number of samples, their location (depth) within the drill hols. More information must be given on the samples used in this study. The information is insufficient. I suggest that the authors add a table in which the samples are described in terms of where in the deposit they are from, the mineralogy, and prevailing textures in each. The samples, or at least a selection thereof, should also be located on cross-sections (Fig. 2). Doing this provides a context for the data that follows.
2- Line numbers are missing from 9 page.
3- It should be useful to designate the table of paragenetic succession of the Fangʼan deposit: mineralogical evolution with time.
4- How do you explain the fact that according to your data «the pyrite is divided into two types: enriched in Fe and depleted in S (S/Fe < 2), and enriched in S and depleted in Fe», while you divide pyrites in the 3 stages?
5- Elements detected by the EMPA method - Co, Ni, As, Cu, Zn are not very characteristic of granites (section 5.4). You indicate in the text that “The Ni content of felsic rocks, such as highly differentiated magmatic rocks (granites), is usually very low. The Ni content in pyrite associated with late granitic hydrothermal fluids is generally very low. In this paper, the Ni content of gold–bearing pyrite in the Fang'an gold deposit was analyzed by EPMA as between 18×10-6 and 1528×10-6”. Do you admit the fact that the source of gold was not granitic magma, but basic, for example?
6- It is necessary to emphasize the initial novelty of your research. It needs more sufficient evidences to prove type of gold deposit. It is not clear from the text that it is this type. Relevant information for an understanding of the ore deposit type must be present. Hydrothermal deposits are also divided into high, medium and low temperature objects; conclusions that are more specific would be desirable, even for coexisting minerals.
7- It makes sense to add data on LA-ICP-MS or S isotopy in pyrite to confirm the conclusions about the genesis and staging of the deposit based on the data obtained.

Round 2
Reviewer 1 Report
According to reviewers' comments and suggestions, authors revised the originial MS, and modified MS have largely been improved in characteristics descriptions, reference data, diccussions as well as English usage. Whereas, some shortcomings are still obvious. For examples, new data presented in this MS are only trace elements of different-stage pyrite which are insufficient for the disscussions and conclusions. In addition, it seems that, excepet the gold occurrence, main discussions and concusions in this MS larely certifies the previou viewpoints such as ore-forming fluid and ore materials origin, ore genesis and associated igneous rocks.
Considering some deficiency and shortcoming that are diffcult to add and modify. The reviewer advise that some additional experiments or new data needed, and this MS is not suitable for publication in Minerals and should be rejected.
Reviewer 3 Report
Dear Authors,
The revised manuscript has clearly responded to my former comments, and some important questions are also explained such us adding missing information about sampling points, their number and places of selection, also you conducted an additional LА-ICP-MS analysis. But there are still some issues that need to be addressed.
In detail, I would suggest
The main problem is obvious the only one sample from the drill hole (ZK1549 and ZK1665) is very small and completely uninformative. It is speculation to draw any conclusions based on these data. At least a few more samples from the drill holes should be included in the sample selection.
1) Line 94. Figure 2 should be completed. You need to add color and symbols, and show the sampling location marked b
2) Line 166. Why in mineral paragenetic sequence of the Fangʼan deposit is missing gold?
